# Population Study Reveals Genetic Variation and Introgression of Four Deciduous Oaks at the Junction between Taihang Mountain and Yanshan Mountain

Ziqi Pei [1,2,†], Qinsong Yang [1,2,†], Xining Chen [1,2], Yu Zong [3], Jinjin Li [1,2], Xiong Yang [1,2], Chenrui Huo [1,2], Yixin Chen [1,2], Na Luo [1,2], Jialu Kang [1,2], Xiaoqian Meng [1,2], Yining Li [1,2], Huirong Zhou [1,2], Jiaxi Wang [1,2], Yong Liu [1,2] and Guolei Li [1,2,*]

1   Research Center of Deciduous Oaks, Beijing Forestry University, Beijing 100083, China
2   Key Laboratory for Silviculture and Conservation, Ministry of Education, Beijing Forestry University, Beijing 100083, China
3   College of Life and Chemistry Sciences, Zhejiang Normal University, Jinhua 321000, China
*   Correspondence: glli226@163.com
†   These authors contributed equally to this work.

**Abstract:** Oaks (*Quercus* spp.) are considered model plants for studying plant evolution and natural gene introgression. Beijing area is at the junction between Taihang Mountain and Yanshan Mountain, and it is an overlapping distribution area of several deciduous oaks native to China. Interspecific hybridization often occurs in sympatric sibling species, resulting in blurred interspecific boundaries and hindering the development of breeding. To have better improvement and protection of these oaks, it is urgently necessary to evaluate the genetic diversity and population structure of these oak species. In this study, we collected eleven populations of four oak species (*Q. variabilis*, *Q. mongolica*, *Q. dentata* and *Q. aliena*) in the Beijing area. By using the polymorphic SSR markers, we analyzed the genetic variation of the collected 400 individuals, investigated the population structure, and found gene introgression events. *Q. variabilis* had a clearer genetic background as compared to the other three species. *Q. mongolica* had a more frequent gene introgression with *Q. dentata* and *Q. aliena*. Our data reveal the introgression and genetic diversity of the four oak species native to the Beijing area, laying the foundation for conducting an oak germplasm nursery with a clear genetic background for protection and further improvement.

**Keywords:** oak; population; genetic variation; SSR; DNA fingerprint

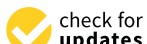

## 1. Introduction

Oaks (*Quercus* spp.) are widespread all over the world. *Quercus* is a large genus in Fagaceae and oaks have been cultivated for a long time, especially in Asia, Europe, and America, for their longevity and renewability [1]. There are over 450 *Quercus* species in the world and 51 in China. Oaks spread over south and north of China. In Beijing, the junction between Taihang Mountain and Yanshan Mountain, several deciduous oaks are natively distributed. *Q. variabilis*, *Q. mogolica*, *Q. dentata*, and *Q. aliena* are the most widely distributed oaks in north of China and have dominated the industrialization and utilization of ecological and economic plants. These four natively oaks are abundant, extensive, and overlapping at the junction between Taihang Mountain and Yanshan Mountain, and some of them are difficult to identify due to interspecific hybridization. This area is the north border for *Q. variabilis* and *Q. aliena* natural distribution and almost the south border for *Q. mongolica* natural distribution in China (Figure S1). Oak species have been formalized into two subgenera: subgenus *Quercus*, comprising *Ponticae*, *Protobalanus*, *Virentes*, *Lobatae* and *Quercus*; and subgenus *Cerris*, encompassing sections *Ilex*, *Cerris*, and *Cyclobalanopsis* [2–4]. Among the native oaks in Beijing area, *Q. variabilis* belongs to *Cerris* section East Asia Cerris with slender leaves, while *Q. mogolica*, *Q. dentata*, and *Q. aliena* belong to *Quercus* section Roburoids with oval and irregular leaves [4].

Oaks being widely spread has been described as "the evolutionary success". One of the explanations of this success is their propensity for hybridization, contributing to adaptive introgression [5], which is increasingly recognized as an important process across diverse lineages of plant [6,7]. Studies have shown that interspecific hybridization and gene introgression frequently happen among species of deciduous oaks. However, the introgression makes it difficult to figure out the differences between two species with close relationship, leading to uncertainty when collecting the germplasm resources, especially for oak species. Studies on gene introgression among *Quercus* species have been carried out for decades. The hybridization and introgression have been clearly found in oak species widely distributed in Europe and North America, such as *Q. rubra*, *Q. suber*, *Q. alba*, and *Q. robur* [7–9]. A study in Japan also demonstrated the interspecific hybridization of *Q. mongolica* and *Q. dentata* [10]. Zeng et al. [11] found that *Q. mongolica* and *Q. liaotungensis* had a high degree of gene introgression in the Beijing area, leading to unclear species morphological characterization. Interspecific hybridization results in gene flow of each species, forming a complex genetic background. The origin of the individual could not be identified. However, the gene introgression among *Q. variabilis*, *Q. mongolica*, *Q. dentata*, and *Q. aliena* is very complex, and the genetic background of these oak species in the Beijing area is still unknown. The ambiguity of genetic background also leads to the inaccuracy of some existing studies in representing a particular oak species.

DNA molecular markers, especially SSR (Simple Sequence Repeat) markers, have been developed for many years and are widely used in recognition of species or cultivars [12–17]. In coconut, 74 polymorphic SSR markers were identified by genotyping-by-sequencing (GBS) analysis of 40 coconut accessions. Different traits could be divided into several groups, which was consistent with the amplicons of SSR markers [18]. Recently, SSR markers have been used in the identification of tree peony varieties, and the 32 phenotypes were divided into three main clusters by 31 SSR markers [19]. In oaks, although SSR markers have been used to characterize the populations worldwide for many years [20–23], genetic resources of native oaks can be found at the junction between Taihang Mountain and Yanshan Mountain, a specific region.

DNA fingerprints can be used to reveal and compare the relationship among individual oak within a species or different species, and can be effectively applied to genetic analysis and breeding [24,25]. We can generate the DNA fingerprint of an individual oak in those areas by using SSR markers to obtain exclusive identity information. In peas, DNA fingerprint and ISSR marker analysis indicated that peas distributed in two places had a tendency of differentiation [26]. Additionally, modern fingerprinting is crucial for varietal protection and germplasm characterization, even for protecting plant breeders' rights. Five microsatellite markers were used to distinguish different varieties of rice, ultimately solving the problem of mixing of two rice cultivars [27]. As there is no oak cultivar native to China, germplasm is very important for plant breeding. High-quality cultivars are the prerequisite for efficient breeding. In order to understand the genetic background and make use of resources for seedling breeding and grafting, it is beneficial to investigate the germplasm resources and obtain the genetic diversity of the four oak species.

Here, we collected 400 individuals of four oak species, including individuals both in natural and artificial forests at the junction between Taihang Mountain and Yanshan Mountain. By using nine SSR markers that were previously developed with high polymorphism, we analyzed the genetic variation among four oak species (*Q. variabilis*, *Q. aliena*, *Q. dentata*, and *Q. mongolica*) and the genetic diversity within one species from different region. By giving fingerprints of these oak individuals, we provided valuable data for further germplasm resource collecting and better improvement and protection of these native oaks. Oak individuals with clear genetic background provide an experimental basis for subsequent molecular breeding and other biological and genetic studies of oaks native to China.

## 2. Materials and Methods

### 2.1. Sample Distribution and Plant Materials

Beijing is located in a region of widespread native oaks (Supplementary Figure S1) [28]. Briefly, in this study, we chose 6 representative regions for sampling in Beijing, which have abundant oak resources (see Supplementary Table S2 for details). The direct distance between any two of the sampling regions is at least 20 km. 10 ha quadrat was investigated in each region, in which oak species without serious pests and diseases were selected. A total of 400 samples were collected in this study, including 136, 74, 77, and 113 individuals from *Q. variabilis*, *Q. aliena*, *Q.dentata*, and *Q. mongolica*, respectively.

Sampling sites are mostly forest parks rich in forest resources. Except for the *Q. variabilis* and *Q. aliena* populations in Jiufeng (JF), other populations are natural populations. Two kinds of populations were selected according to the previous speculation that the genetic diversity was low in artificial populations and high in natural populations. These areas are the north border of the North China Plain and belong to the warm temperate semi-humid continental monsoon climate zone, with four distinct seasons and abundant rainfalls.

According to the earlier investigation on the spot, all individuals in good growth condition were labeled and numbered. After photographing and recording the position, we collected young leaves or dormant buds. All the samples were frozen in liquid nitrogen and stored at −80 °C until they were used for DNA extraction and isolation. Numbers of samples in each region are shown in Supplementary Table S2.

### 2.2. DNA Isolation and SSR Amplification

Genomic DNA of 400 samples were extracted by using Plant DNA Isolation Mini Kit (Vazyme, Nanjing, China). User manual and modified cetyltrimethyl ammonium bromide method was also used, as previously described [29,30]. The DNA was eluted into 50 μL Tris-EDTA buffer as stock solution. Subsequently, the quality and the concentration of the genomic DNA were quantified using 1% agarose gel electrophoresis and a NanoDrop 2000 spectrophotometer (Thermo Scientific, Waltham, MA, USA), respectively. The genomic DNA was diluted and separated to 20 ng/μL aliquots and stored at −20 °C for polymerase chain reactions (PCR) amplification. Nine polymorphic SSR primer pairs were selected as previously reported [31], namely QrZAG96, QrZAG102, QrZAG112, QrZAG7, Qden03011, Qden03021, Qden03032, Qden05011, and Qden05031. The information of primers is listed in Supplementary Table S3.

The nine pairs of verified SSR primers with M13 (-21) tail at their 5′ end were used for PCR amplification, together with M13 (-21) tailed fluorescent primers (M13-FAM, M13-ROX, M13-HEX) as previously described [30,32]. PCRs were conducted in 15 μL reaction volumes containing 1 μL of genomic DNA (20 ng/μL), 7.5 μL of 2× Rapid Taq Master Mix (Vazyme), 0.4 μL of fluorescent primer, 0.1 μL of forward SSR primer, 0.5 μL of reverse primer, and 5.5 μL of ddH$_2$O. PCR procedures were as follows: 95 °C (5 min), 32 cycles at 95 °C (30 s)/55–59 °C (30 s) (depending on the Tm value of SSR primers)/72 °C (30 s), followed by 8 cycles 95 °C (30 s)/53 °C (30 s)/72 °C (15 s) and a final extension at 72 °C for 5 min. PCR products were sent to Ruibo BioTech (Beijing, China) for capillary fluorescence electrophoresis detection to read the length of the PCR products.

### 2.3. Data Analysis

2.3.1. Characteristics and Genetic Diversity of SSR Primers

GeneMarker 2.2.0 software was used to read the polymorphic loci and obtain the genotypes of all samples. Linkage disequilibrium analysis was tested for all locus pairs in each population by randomization using Arlequin 3.1 [33]. Polymorphism parameters of 9 pairs of SSR markers were calculated by GenAlEx6.5 [34], respectively, by the number of alleles (Na), effective alleles (Ne), Shannon's information index (I), observed heterozygosity (Ho), expected heterozygosity (He), Fixation Index (F), and so on.

2.3.2. Genetic Differentiation among Oak Populations and Optimal Population Classification

The analysis of the population structure was assessed in STRUCTURE 2.3.4 [35] using a Bayesian clustering approach setting parameters with a burn-in period of 100,000 iterations and 100,000 MCMC iterations after burn-in. In this study, we used "admixture model" and "allele frequencies correlated" during the modeling process. This approach revealed genetic structure by assigning individuals or predefined groups to K clusters. Different K values that ranged from 1 to 10 were used to infer the number of clusters for 10 replicate runs. ΔK (ΔK = mean (|L"(K)|)/sd[L(K)]) and lnPPK were shown in STRUCUTRE Harvester [36], using the results of STRUCTURE.

Different results could be produced in the same conditions by different 10 replicate cluster analyses of the same data. Thus, CLUMPP 1.1.2 [37] and R 4.2.0 were used to average these 10 replicate results generated from STRUCTURE data and visualize the outcomes separately. Proportion of gene pools distributed in 10 oaks populations were plotted on maps in the form of pie charts. Depending on the specific coordinates of the sampling sites, pie charts of different populations were plotted on the topographic map. The clustering tree of both individuals (NJ) and populations (UPGMA) were made using PowerMarker 3.25 [38]. Trees were then colored and modified using Adobe Illustrator CC 2018.

2.3.3. Genetic Diversity of 11 Oak Population and PCoA Analysis

Na, Ne, Ho, He, and fixation index (F), Fst, and gene flow (Nm) among 11 populations were calculated by GenAlEx6.5. The software was used to conduct the PCoA of the four oak species and analyze the length range of the amplified alleles among different species. According to the length range of the amplified alleles, the histogram was shown using SigmaPlot 12.5.

## 3. Results

### 3.1. Polymorphism of the Chosen SSR Primers

As previous studies have validated that the nine pairs of SSR markers have high polymorphism and stability and identify oak species [39,40], we chose these SSR markers to identify the genetic background of the four oak species (*Q. variabilis*, *Q. mongolica*, *Q. dentata*, and *Q. aliena*). Meanwhile, the results of LD analysis showed that nine locus were in linkage equilibrium. First, we collected 400 individual samples (including 136 of *Q. variabilis*, 113 of *Q. mongolica*, 77 of *Q. dentata*, and 74 of *Q. aliena*) from six regions of Beijing (see Supplementary Table S2 for details). We used these 400 samples to validate the polymorphism of the nine SSR markers. The nine SSR markers amplified 215 alleles in total, with an average of 23.9 alleles per locus (Supplementary Table S1). The observed heterozygosity (Ho) ranged from 0.45 to 0.82, with an average of 0.61. For all the observed loci, the Ho value was lower than expected (He), which ranged from 0.81 to 0.96 with an average of 0.87 (Supplementary Table S1).

### 3.2. Genetic Variation within Oak Species and Populations

We further analyzed the genetic variation of populations of different oak species from different regions by using SSR amplification results. Overall, *Q. variabilis* has the lowest Ho and He value among the four species, while *Q. mongolica* has the highest Ho and He value (Table 1). In *Q. variabilis*, the number of alleles per locus (Na) ranged from 6.56 to 10.89 among different populations in different regions, and the average Na was 8.61. *Q. variabilis* populations on Dayang Mountain (DY-Qv) and Zhoukoudian (ZK-Qv) have lower Ho compared to those on Jiufeng (JF-Qv) and Shangfang Mountain (SF-Qv), with the highest inbreeding coefficient (F = 0.26 and 0.31).

**Table 1.** Genetic variation of different oak populations.

| Species | Populations [1] | Na | Ne | Ho | He | F |
|---|---|---|---|---|---|---|
| *Q. variabilis* | JF-Qv | 6.89 | 4.19 | 0.59 | 0.66 | 0.04 |
| | ZK-Qv | 10.11 | 4.92 | 0.50 | 0.71 | 0.31 |
| | SF-Qv | 6.56 | 4.59 | 0.56 | 0.67 | 0.18 |
| | DY-Qv | 10.89 | 4.64 | 0.49 | 0.69 | 0.26 |
| | Qv-mean | 8.61 | 4.59 | 0.54 | 0.68 | 0.20 |
| *Q. aliena* | JF-Qa | 10.11 | 5.90 | 0.59 | 0.73 | 0.19 |
| | SF-Qa | 12.11 | 6.49 | 0.61 | 0.76 | 0.21 |
| | Qa-mean | 11.11 | 6.19 | 0.60 | 0.75 | 0.20 |
| *Q. dentata* | SF-Qd | 6.22 | 4.77 | 0.61 | 0.76 | 0.18 |
| | ZK-Qd | 3.33 | 2.78 | 0.52 | 0.59 | 0.07 |
| | DY-Qd | 13.11 | 6.19 | 0.69 | 0.81 | 0.14 |
| | Qd-mean | 7.56 | 4.58 | 0.61 | 0.72 | 0.13 |
| *Q. mongolica* | YM-Qm | 16.89 | 8.28 | 0.67 | 0.86 | 0.22 |
| | BH-Qm | 12.67 | 8.15 | 0.74 | 0.86 | 0.14 |
| | Qm-mean | 14.78 | 8.21 | 0.70 | 0.86 | 0.18 |

[1] The name of population is made up of regions and species, e.g., JF-Qv means *Q. variabilis* samples in Jiufeng.

In the SF-Qa population, Na, Ne, Ho, and He values were higher than those in JF-Qa population, while they have a similar sample amount (35 and 39 in SF and JF, respectively, Supplementary Table S2). The inbreeding coefficient of SF-Qa population is lower than that in JF-Qa population (Table 1), suggesting the higher genetic diversity of *Q. aliena* populations in Shangfang Mountain where natural populations are distributed. Similar patterns have been found in populations of *Q. variabilis*. Therefore, this result indicated that the genetic diversity of the artificial population was lower than that of the natural population.

For *Q. mongolica*, BH-Qm population has a higher Ho value and much lower inbreeding coefficient as compared to YM-Qm population, indicating that *Q. mongolica* in Baihua Mountain has abundant genetic diversity.

Next, we calculated the genetic differentiation coefficient (Fst) and gene flow (Nm) among populations within the same oak species in different region. The results showed that *Q. dentata* had the highest genetic differentiation coefficient and lowest gene flow among populations from Shangfang Mountain, Dayang Mountain, and Zhoukoudian (Table 2). *Q. mongolica* had the lowest Fst (0.01), indicating that only 1% variation happened between YM-Qm population and BH-Qm population.

**Table 2.** Genetic variation among populations within the same oak species in different regions.

| Species | Regions | Fst [1] | Nm |
|---|---|---|---|
| *Q. variabilis* | JF SF DY ZK | 0.04 | 6.16 |
| *Q. aliena* | JF SF | 0.02 | 12.25 |
| *Q. dentata* | SF DY ZK | 0.08 | 2.80 |
| *Q. mongolica* | YM BH | 0.01 | 17.61 |

[1] Fst and Nm are the parameters of genetic variation. Fst: genetic differentiation coefficient. Nm: gene flow.

### 3.3. Population Structure and Gene Introgression

A neighbor-joining tree was conducted to show the genetic relationship among all the observed individuals. Most of the individuals of *Q. variabilis* had the longest genetic distance with the other three species. Notably, a few individuals of *Q. variabilis* were grouped with *Q. dentata* and *Q. mongolica* (Figure 1), indicating that possible gene introgression happened among species. Meanwhile, *Q. dentata*, *Q. aliena*, and *Q. mongolica* had more individuals grouped with each other, which suggested that gene introgression happened more frequently among these three species, confirming that these three species had a closer relationship as compared to *Q. variabilis*.

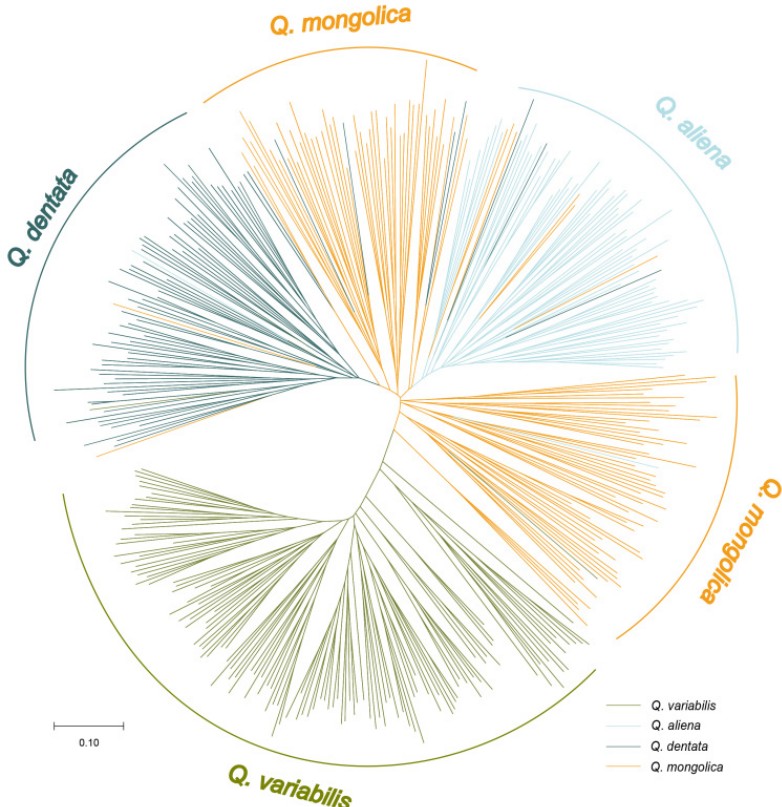

**Figure 1.** Neighbor-joining tree of 400 oak individuals from four species. Different colors indicate different species. The cluster of *Q. variabilis* (green), *Q. aliena* (blue), *Q. dentata* (cyan), and *Q. mongolica* (orange) is shown in the circle NJ tree. Length of lines show the genetic distance. The scale bar represents 0.1 genetic distance.

The population structure of 400 oak individuals was also confirmed by principal coordinates analysis (PCoA). We firstly used all the samples to do the PcoA analysis. The results showed that the first coordinate separated *Q. variabilis* and the other three species, while the other three species were grouped together (Figure 2a). In addition, PCoA analysis by using the samples for each species in different regions could not generate the significant group (Supplementary Figure S2). The results show that there was no distinct genetic differentiation among the different populations from different regions in the Beijing area for each oak species.

The 400 oak individuals were evaluated for population stratification. The assumed number of clusters was set from *K* = 1 to 10. The Bayesian analysis by STRUCUTRE Harvester showed that the individuals were suitable to be divided into four groups (*K* = 4) (Supplementary Figure S3). The STRUCTURE results showed that *Q. variabilis* (green) has a relatively clear genetic background (Figure 2b). Further, four native oaks species were evaluated for population stratification, respectively. The assumed number of clusters

was set from *K* = 2 to 5 [41]. The STRUCTURE results showed that each species has a low degree of differentiation among different sampling region, especially *Q. dentata* and *Q.mongolica* (Supplementary Figure S4), with high Nm values (Table 2). This result was consistent with PCoA analysis and phylogenetic tree (Figures 1, 2 and S2). *Q. mongolica* had abundant genetic diversities and gene introgression individuals from *Q. aliena* and *Q. dentata*, indicating that *Q. mongolica*, had a closer genetic relationship with *Q. aliena* and *Q. dentata* as compared to *Q. variabilis*, as we could also see the population structure when *K* = 2 and *K* = 3 (Supplementary Figure S5).

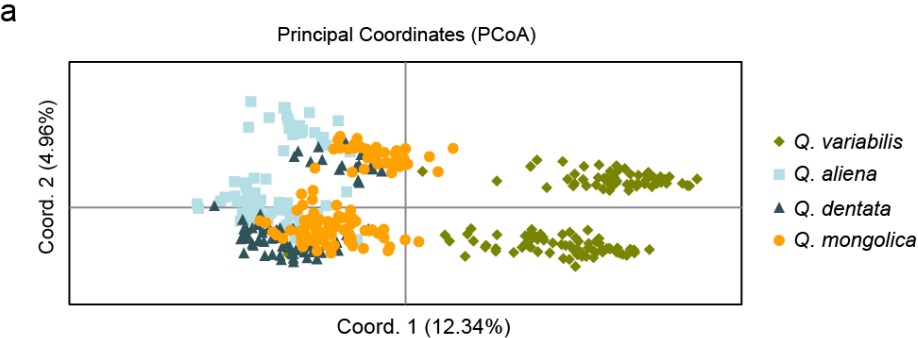

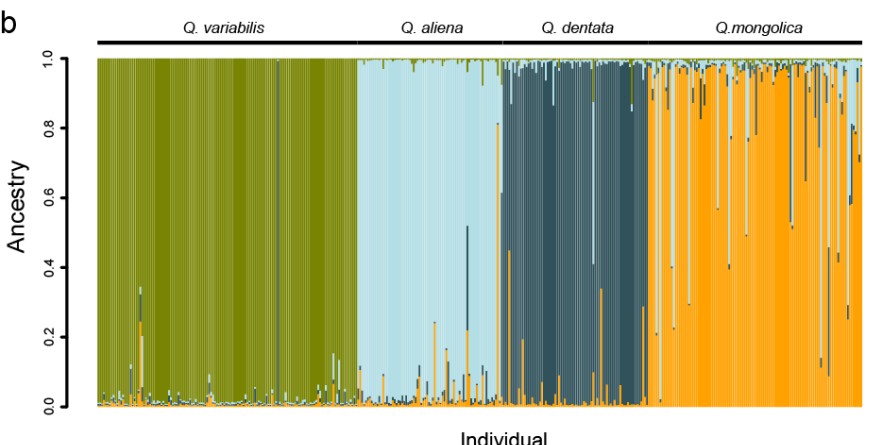

**Figure 2.** PCoA and population structure for all the 400 individuals. (**a**) PCoA groups of four oak species are calculated and plotted by GenAlEx6.5. Different colors indicate different species, and each point represents an individual. (**b**) Population structure based on Bayesian clustering approaches with *K* = 4 by STRUCTURE software. Each bar represents a single individual and different colors indicate different subpopulations. The individual species are shown at the top of the bars.

The regions where we collected samples are located on the edge of the North China Plain (Figure 3), and they are near the borders of natural distribution of *Q. mongolica* and *Q. aliena* (Supplementary Figures S1 and S6). The pie charts indicated a proportion of gene pools distributed in 10 oaks populations, of which *Q. mongolica* is the most complex.

We then used the unweighted pair-group method and the arithmetic (UPGMA) algorithm to build the phylogenetic tree of populations of four species from different regions in the Beijing area. Populations from one species were grouped together and *Q. mongolica* had a closer genetic relationship with *Q. aliena* than that with *Q. dentata* (Figure 4). Notably, JF-Qv population, which was an artificial population, had a further genetic distance with other three *Q. variabilis* populations, indicating that the individuals might be transplanted from another area outside Beijing.

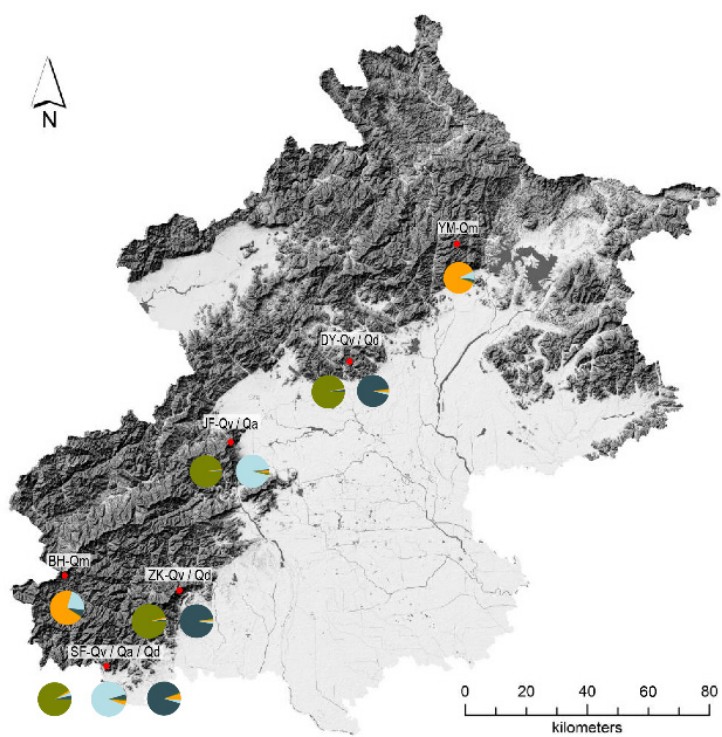

**Figure 3.** Population distribution map and genetic background of all samples. Different colors indicate the different genetic clusters in STRUCTURE analysis.

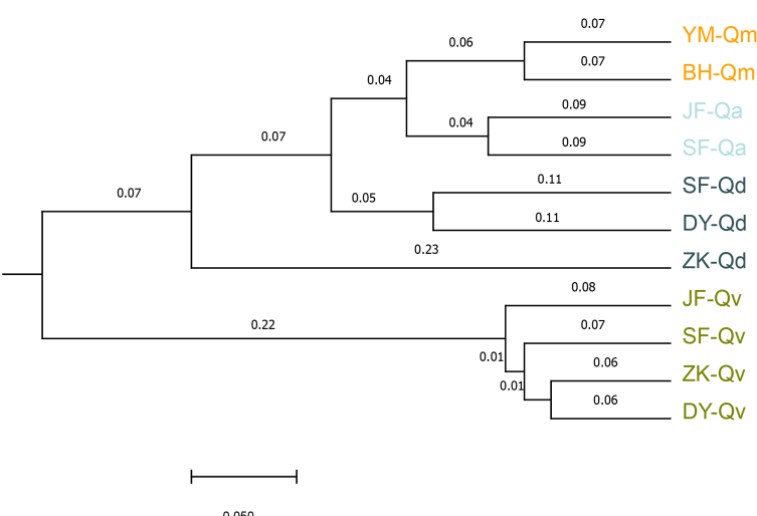

**Figure 4.** The UPGMA analysis of populations from different regions. Eleven populations mentioned above are evaluated. The scale bar represents 0.05 genetic variation. The abbreviations of the population names are listed in Table S2.

### 3.4. DNA Fingerprints of Oak Species in Beijing Area

As there was no oak germplasm nursery with clear genetic background in China, we provided the fingerprints of the observed 400 oak individuals. The DNA fingerprints of these 400 individuals were represented with numbers that showed the allele length and letters that indicated the amplified SSR alleles (Supplementary Table S4). The length of amplified alleles varied among different species. Qden03021 had a large range of amplicons in *Q. variabilis* as compared to the other species, while Qden05031 had a large range of amplicons in *Q. mongolica* (Supplementary Figure S7), suggesting the disparities in polymorphism of different alleles in different oak species.

## 4. Discussion

Oaks are valuable in ecologic, economic, and research fields. They have proved to be a successfully evolved species in the world, with extensive radiation and expansion [5], and are ideal materials for us to study gene introgression and natural hybridization of woody plants [7,21,42]. Hipp et al. [3] used genotyping-by-sequencing and SNP markers to generate large-scale data, demonstrating frequent gene diversity and introgression of oak species in *Quercus* genus. We conducted a fine-scale study to reveal the genetic relationship of oak species at the junction between Taihang Mountain and Yanshan Mountain, taking a close look at the genetic diversity within and among species, and also observed the gene introgression among these oak species. We collected 400 individuals of oak species in the area which is the north border of North China Plain. In this area, several native oak species, including *Q. variabilis*, *Q. aliena*, *Q. dentata*, and *Q. mongolica* are naturally distributed. As there is no improved cultivar for any of these oak species native to China, it is meaningful to detect the genetic variation of these individuals, preparing for germplasm nursery using accessions with a clear genetic background.

Previous studies have screened some SSR markers with high polymorphism for oak species [31,43]. We used these SSR markers and further confirmed the polymorphism of the picked 9 markers. For the individuals of the four oak species in Beijing area, these SSR markers exhibit high polymorphism, and 215 alleles were detected in total, 23.9 on average (Supplementary Table S1), much higher than the studies in *Q. petraea* [12] and similar to the studies on *Q. mongolica* and *Q. liaotungensis*, which collected 1166 individuals [11]. This result indicates that these SSR markers are efficient for oak species identification [43]. Range size shows some relationship to heterozygosity and allelic richness [44], in that *Q. mongolica* had the highest heterozygosity (He = 0.86) and allelic richness (Na = 14.78), suggesting that *Q. mongolica* had the most widespread distribution among the four species. Naturally, the phylogenetically close relationship between *Q. dentata* and *Q. aliena* with *Q. mongolica* could be well explained. Whether they originated and evolved in *Q. mongolica* remains unconfirmed. Both total alleles and expected heterozygosity (He) were lower than the studies in *Q. variabilis* on a national scale [44], speculating that it might due to artificial population. It is interesting that the inbreeding coefficient (F) of the artificially planted *Q. variabilis* population (JF-Qv) was 0.04 (Table 1), suggesting that the individuals probably came from random mating populations. Meanwhile, the UPGMA analysis showed that JF-Qv artificial population had a phylogenetically distant relationship with other *Q. variabilis* populations in this area (Figure 4), indicating that the trees in Jiufeng might be transplanted from other areas outside Beijing. Thus, the sampled individuals might come from a different provenance. In contrast, the other artificial population, *Q. aliena* population in Jiufeng (JF-Qa), had a strong gene flow (Nm = 12.25) with the natural population in Shangfang Mountain (SF-Qa) (Table 2), suggesting that some individuals in JF-Qa population might be transplanted from Shangfang Mountain. Gene flow determines the genetic structure and survival potential of future populations of a species [45], and the previous study reported a lower gene flow (Nm = 3.648) among the population of *Q. variabilis* than our study [46], which objectively reflected that *Q. variabilis* distributed at the junction between Taihang Mountain and Yanshan Mountain had more survival potential. In addition, SF-Qa natural population has more abundant genetic diversities than JF-Qa artificial population (Table 1), further proving that JF-Qa might be generated from the same provenance.

The phylogenetic tree, combining UPGMA analysis and STRUCTURE analysis, showed that *Q. variabilis* evolved separately from the other three species (Figures 1, 2 and 4). This finding is consistent with previous studies using SNP markers and limited individuals within one species [3]. As Hipp et al. [3] reported, *Q. mongolica*, *Q. aliena*, and *Q. dentata* belong to *Quercus* section *Roburoids*, while *Q. variabilis* belongs to *Cerris* section *East Asia Cerris*, together with *Q. accutisima*, and *Q. dentata* evolves separately from *Q. mongolica* and *Q. aliena*. The introgression of *Q. variabilis* and *Q. accutisima* has already been confirmed by using high-quality genomic resources [47]. Our results also demonstrate that *Q. mongolica* and *Q. aliena* have a closer genetic relationship as compared to *Q. dentata*

(Figures 1, 2 and 4). The gene flow between BH-Qm and YM-Qm reached 17.61, much higher than the other three species. Considering the geographical distance between Baihua Mountain and Yunmeng Mountain, the two *Q. mongolica* populations might have evolved together and separated recently. As many previous studies have demonstrated that gene introgression happens frequently in *Quercus* genus [7,9,21,48], it could not be ignored when analyzing the oak population in Beijing area. The STRUCTURE analysis and the phylogenetic tree showed that gene introgression individuals were grouped with corresponding species. *Q. mongolica* has the most individuals that have gene introgression with *Q. dentata* and *Q. aliena* (Figures 1 and 2). *Q. variabilis* has the least gene introgression individuals, suggesting that *Q. variabilis* has a phylogenetically distant relationship with the other three species. A possible contributor is the lower successful rate of interspecific hybridization with the other three species in natural conditions. This result provides guidance for distant hybridization breeding of *Q. variabilis*, which should take the hybridization compatibility into consideration. In addition, as previously reported, *Q. variabilis* has a distinct flowering time compared to the other three species [49], which might also create less chances for interspecific hybridization [50]. Previous studies on *Q. dentata* and *Q. aliena* showed that the male flowers of *Q. dentata* and the female flowers of *Q. aliena* in this area had a certain overlap [51], suggesting that these two oaks had a frequent outcrossing.

The junction between Taihang Mountain and Yanshan Mountain is not only near the north border of the natural distribution of *Q. variabilis* and *Q. aliena* but also near the south border of the natural distribution of *Q. mongolica*. *Q. mongolica* distributed on the mountains with an altitude over 600 m. The north part of this distribution area is high while the south part is low, topographically, with a warm temperate sub-humid continental monsoon climate. Considering the pollen dispersal of oak species and the northwest wind direction in spring at the junction between Taihang Mountain and Yanshan Mountain, it could be explained that southwest of Beijing had abundant species diversities of deciduous oaks, which may also explain the warmer climate (Figure 3). Fine-scale spatial genetic structure of the oak species could be further studied in the Beijing area to further identify the interspecific hybrids and their adaption to the environment [40,52,53]. During our investigation on the spot, we observed that the leaf shape of *Q. dentata* was similar to *Q. mongolica* (Supplementary Figure S8) but quite different from the leaf of *Q. variabilis*. Further studies could use SNP markers to link the morphological traits or environmental adaptions (e.g., temperature, drought, salt, etc.), and the genotype and more oak species native to China should be identified by molecular markers. With a clearly identified genetic background, we could use these genetic resources for breeding.

DNA fingerprints could give the putative cultivar an identity, which is very important for intellectual property protection. Here, we provided all the DNA fingerprints of 400 samples generated from SSR data (Supplementary Table S4). Further study could be done to get the link between DNA fingerprints and their morphological traits by using more SSR markers or SNP markers like studies in some other species [18,19], making it clearer to focus on some specific trait breeding.

In conclusion, we detected the genetic variation of natural and artificial populations of four oak species native to China and confirmed the genetic relationship and gene introgression of *Q. variabilis*, *Q. mongolica*, *Q. aliena*, and *Q. dentata*. Our work has laid a foundation for collecting oak germplasm and provides valuable data for further protection and improvement. In addition, we screened some individuals with a clear and simple genetic background for whole genome sequencing and cross-breeding.

**Supplementary Materials:** The following supporting information can be downloaded at: https://www.mdpi.com/article/10.3390/f13101647/s1, Figure S1: Distribution map of *Q. variabilis* (a), *Q. aliena* (b), *Q. dentata* (c) and *Q. mongolica* (d) respectively in China. The red squares represent Beijing area. This figure is modified from Wang et al. [1]. Figure S2: PCoA analysis of different populations within the same species. Different color and shape of the dots indicate the different populations. Figure S3: ΔK and lnPPK values are showed with blue dot. The values are calculated according to *K* = 1 to 10. Figure S4: STRUCTURE analysis of 4 native oaks when *K* = 2 to 5. *Q. variabilis* (a),

*Q. aliena* (b), *Q. dentata* (c) and *Q. mongolica* (d) respectively. Figure S5: STRUCTURE analysis of populations when K = 2 and K = 3. Dark blue background shows *Q. variabilis* cluster. Figure S6: The sampling regions of four native oaks. *Q. variabilis* (a), *Q. aliena* (b), *Q. dentata* (c) and *Q. mongolica* (d) respectively. Figure S7: Length variety of the amplicons of each allele in different oak species. Green, blue, cyan and orange bars indicate length variety of each allele in *Q. variabilis*, *Q. aliena*, *Q. dentata* and *Q. mongolica* respectively. Figure S8: Leaf surface and back of *Q. variabilis* (a), *Q. aliena* (b), *Q. dentata* (c) and *Q. mongolica* (d) respectively. Table S1: Main genetic parameters for the chosen 9 SSR primers. Table S2: Sample information. Table S3: Information of the 9 SSR primers. Table S4: DNA fingerprints of 400 samples.

**Author Contributions:** Q.Y. and G.L. designed the experiments. Z.P. and Q.Y. conducted most of the experiment, analyzed the data. Z.P. and Q.Y. wrote the manuscript. Z.P., X.C. and J.L. conducted DNA extraction and SSR PCR experiment. Y.Z. analyzed the data and revised the manuscript. Z.P., Q.Y., X.C., J.L., X.Y., C.H., Y.C., N.L., J.K., X.M., Y.L. (Yining Li), H.Z. and J.W. collected the samples. Y.L. (Yong Liu) and G.L. supervised the project and revised the manuscript. All authors have read and agreed to the published version of the manuscript.

**Funding:** This work was supported by the Fundamental Research Funds for the Central Universities (2021ZY11), the China Postdoctoral Science Foundation (2021M700451).

**Data Availability Statement:** The data presented in this study are available on reasonable request from the corresponding author.

**Acknowledgments:** We thank Yan Wang for providing useful suggestions on SSR data analysis and manuscript preparing. We thank Kundong He for English language improvement.

**Conflicts of Interest:** The authors declare no conflict of interest.

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
