# Peer review of "Population Study Reveals Genetic Variation and Introgression of Four Deciduous Oaks at the Junction between Taihang Mountain and Yanshan Mountain"

_forests, doi:10.3390/f13101647_

Round 1

Reviewer 1 Report

See attached PDF.

Reviewer 2 Report

I began to read the submitted manuscript with great interest, as it deals with very interesting oak species in China and topics such as hybridization and breeding. However, even in the introduction, it became less and less clear to me what the focus of this work is. I missed the goal and objectives, and the description of how to achieve them (both in the introduction). 

There are many points related to the unclear design that make it questionable whether the results make sense to summarize anything. Here are some of the points that are not clear to me:

Why did the authors choose these four species?

Why did they decide to focus on the area between Taihang Mountain and Yanshan Mountain?

Why did they choose natural and artificial populations?

Why did they choose different numbers of populations per species?

Why is the population size so different, ranging from 3 to 68?

Is the selected marker set sufficient to study introgression in these oak species?

A map showing the distribution of all species in China and the area studied may be very helpful to understand some points of the study design.

The introduction would additionally benefit from explanations such as: 

Why tree cultivars and tree germplasm nurseries are important, and also how individuals are selected for them (in the introduction).

What does that  mean, complex genetic background?

Are nuSSR appropriate to explain phylogenetic relationships between species (this is new to me).

How and why did they select 10 individuals per oak species?

Although the topics the authors address here are well suited for publication in this special issue of the Journal Forests and are of interest to readers, the current status of the paper does not meet the first and most important requirement for publication, which is the clear focus of the paper supported by the scientific literature. 

I therefore propose to ask the authors for a comprehensive revision.

Round 2

Reviewer 2 Report

Dear authors, 

In my first review, I asked 5 "WHY" questions, which you answered adequately in your revised version. In the following part, I ask you to make additional corrections (minor changes) that will further improve your manuscript and thus the readers' understanding. These are:

Line 23; replace native to Beijing by native to Beijing area

Lines 70-72; please remove the sentence “In Alium…..natural populations. This sentence is not relevant.

Lines 72-74; The sentence is not finished

Lines 95-97; The sentence doesn’t make sense in English, please correct.

Line 95; Please explain more exact what do you mean here by clear and simple

Line 102; …between the two… which two do you mean here?

Line 174; Have these microsatellites also been proven to be useful for studying introgression?

Table 1: This table can be moved to the Supplementary Material

Table 3: please put the entire name of the species such as Q.variabilis instead of abbreviation only

Lines 222-224; These you can tell without additional dating or demographic analyses 

Lines 226-227; How can authors be sure that there was rather misidentification of species instead of possible gene introgression?

Figure 1: Please also add 1b, where you will mark the different populations by different colors? 

Figure Supplementary 3: Please use this Figure in the main manuscript, In an additional Figure, zoom in into the study area with 6 regions and the mountain ranged where you collected your populations.

Figure Supplementary 2: Please enlarge the size of all four PCoA (A4 Format?) that a reader can recognize the different individuals of each population better?

Figure 2a: How the authors explain the gap revealed by coordinate 2?

Figure 2b: please do run additional STRUCTURE for each species (as assigned in Fig. 2b) separately. See also publications on European oaks of Neophytou et al. 2010, 2014, 2015

Page 304; please replace the word characterization by identification.

Page 373; Here we select 10 individuals... Not really, in the Supplementary Table 3, you listed all 400.
